# COMBINE AND CONQUER: A META-ANALYSIS ON DATA SHIFT AND OUT-OF-DISTRIBUTION DETECTION

## ABSTRACT

This paper presents a universal method for integrating detectors and evaluates various approaches for addressing data distribution shifts and detecting out-of-distribution data. We achieve this by normalizing detector scores into p-values using quantile normalization, effectively transforming the problem into a multivariate hypothesis test. We then combine these tests using established meta-analysis tools, resulting in a more effective detector with consolidated decision boundaries. Additionally, we can create a fully interpretable criterion by adjusting the final statistics of the in-distribution scores. Our framework is highly adaptable for future developments in detection scores. Through a meticulous empirical investigation, we analyze different types of shifts with varying degrees of impact on data, demonstrating that our approach significantly enhances overall robustness and performance across various domains, shift types, and out-of-distribution detection scenarios.

## 1 INTRODUCTION

Deploying AI systems in real-world applications is not without its challenges. Although these systems are evaluated in static scenarios, in practice, they encounter a dynamic and evolving environment. One of the most pressing issues is preventing and reacting to *data shift* (Quionero-Candela et al., 2009). It occurs when the data distribution used to train an AI model no longer matches the data required to process. It can happen gradually or suddenly and can be caused by various factors, e.g., changes in user behavior or degradation in operating conditions, which can have severe consequences in safety-critical applications (Amodei et al., 2016) such as autonomous vehicle control (Bojarski et al., 2016) and medical diagnosis (Subbaswamy & Saria, 2020).

As modern machine learning models can be difficult and expensive to adapt, an appropriate detection of drifts may reduce the need for retraining. Even though shifts in distributions can result in significant performance declines, in reality, distributions also undergo shifts that are harmless (Gemaque et al., 2020). As a result, professionals should focus on discerning detrimental shifts that harm predictive performance from unimportant shifts that have little impact. In other words, detecting harmful drifts may lead to a discriminating method to decide when retraining is necessary.

This paper explores ways to improve the *detection* of performance-degrading shifts by ensembling existing detectors in an unsupervised manner. Each detector can be formalized as a test of equivalence of the source distribution (from which training data is sampled) and target distribution (from which real-world data is sampled) through the lens of a predictive model. Our approach is motivated by the fact that different detection algorithms may make trivial mistakes in different parts of the data space without any assumptions on the test data distribution (Birnbaum, 1954). The challenge is to develop a widely applicable method for combining detectors to alleviate catastrophic errors.

We make the following **contributions**:

1. A simple and convenient ensembling algorithm for existing detectors leading to better generalizability by incorporating effects that may not be apparent in individual detectors;

2. A framework to adapt any single example detector to a window-based data shift detector;

3. A comprehensive empirical validation encompassing single example out-of-distribution detection and window-based data distribution shift detection.

## 2    RELATED WORKS

**Window-based data shift detection.** This line of work proposes methods for detecting shifts in data distribution using multiple samples. Lipton et al. (2018) presents a technique for detecting prior probability shift. Rabanser et al. (2019) studies two-sample tests with high dimensional inputs through dimensionality reduction techniques from the input space to a projected space. Cobb & Looveren (2022) explores two sample conditional distributional shift detection based on maximum conditional mean discrepancies to segment relevant contexts in which data drift is diminishing.

**Misclassification detection.** Misclassification detection aims to reject in-distribution samples misclassified in test time with roots in rejection option (Chow, 1957) and uncertainty quantification (Abdar et al., 2021). A natural baseline is the classification model's maximum softmax output (Hendrycks & Gimpel, 2017; Geifman & El-Yaniv, 2017). Granese et al. (2021) introduce a simple framework that considers the entire probability vector output. Gal & Ghahramani (2016); Lakshminarayanan et al. (2016) are popular approaches for estimating uncertainty from a Bayesian inference perspective. Even though this line of work focuses mainly on detecting problematic in-distribution samples while we focus on distributional drifts, our framework could be extended to it.

**Novelty and out-of-distribution detection.** Out-of-distribution (OOD) detection is also referred to xtin the literature as open-set recognition (Geng et al., 2021), one-class novelty detection (Pimentel et al., 2014), or semantic anomaly detection (Wang et al., 2020). Haroush et al. (2022) also frames OOD detection as a statistical hypothesis testing problem and aggregates p-values on multiple layers channels of the network in a hierarchical fashion. Their final method relies heavily on the architecture of convolutional neural networks, reduction functions, and they do not adjust for correlation between the test statistics as they point out in Section 4.2 therein. Overall, methods are taxonomized into confidence-based Hein et al. (2019); Hendrycks & Gimpel (2017); Liang et al. (2018); Hsu et al. (2020); Liu et al. (2020); Hendrycks et al. (2022); Sun & Li (2022), which rely on the logits and softmax outputs; feature-based (Sastry & Oore, 2020; Quintanilha et al., 2019; Sun et al., 2021; Huang et al., 2021; Zhu et al., 2022; Colombo et al., 2022; Dong et al., 2021; Sun et al., 2022a; Song et al., 2022; Lin et al., 2021; Djurisic et al., 2023; Lee et al., 2018; Ren et al., 2021; Sun et al., 2022b), which explores latent representations; mixed feature-logits (Gomes et al., 2022; Wang et al., 2022); training, likelihood estimation and reconstruction based (Schlegl et al., 2017; Vernekar et al., 2019; Xiao et al., 2020; Ren et al., 2019; Zhang et al., 2021; Kirichenko et al., 2020) methods. We consider these methods to be complementary to our work as they focus on developing single discriminative OOD scores. By analyzing the results from a recent benchmark (Zhang et al., 2023), it is evident that there is no single winner, which empirically motivates this work.

## 3    METHODOLOGY

This section digs into the methodology for detecting distribution shifts in data streams inputted to deep neural networks. We define data stream in Section 3.1, we recall the various types of shifts in Section 3.2, and we formalize single sample and window-based detection in Section 3.3.

### 3.1    BACKGROUND

Let $\mathcal{X} \subseteq \mathbb{R}^d$ be a continuous feature space, and let $\mathcal{Y} = \{1, \ldots, C\}$ denote the label space related to some task of interest. We denote by $p_{XY}$ and $q_{XY}$ the underlying source and target probability density functions (pdf) associated with the distributions $P$ and $Q$ on $\mathcal{X} \times \mathcal{Y}$, respectively. We assume that a machine learning model $f : \mathcal{X} \rightarrow \mathcal{Y}$ is trained on some training set $\mathcal{D}_n = \{(\boldsymbol{x}_1, y_1), \ldots, (\boldsymbol{x}_n, y_n)\} \sim p_{XY}$, which yields a model that, given an input $\boldsymbol{x} \in \mathcal{X}$, outputs a prediction on $\mathcal{Y}$, i.e., $f(\boldsymbol{x}) = \arg\max_{y \in \mathcal{Y}} p_{\hat{Y}|X}(y \mid \boldsymbol{x})$. At test time, an unlabeled sequence of inputs or *data stream* is expected, sampled from the marginal target distribution $q_X$.

**Definition 3.1** (Data stream). A data stream $\mathcal{S}$ is a finite or infinite sequence of not necessarily independent observations typically grouped into *windows* (i.e., sets $\mathcal{W}_j^m = \{x_j, \ldots, x_{j+m-1}\} \sim q_X$) of same size $m$,

$$\mathcal{S} = \{\boldsymbol{x}_1, \ldots, \boldsymbol{x}_m, \ldots\} = \bigcup_{j=1}^{\infty} \mathcal{W}_j^m. \tag{1}$$

## 3.2 DATA-SHIFT

In real-world applications, data streams usually suffer from a well-studied phenomenon known as *data distribution shift*[1] (or data shift for short). Data shift occurs when the test data joint probability distribution differs from the distribution a model expects, i.e., $p_{XY}(\boldsymbol{x}, y) \neq q_{XY}(\boldsymbol{x}, y)$. Due to this mismatch, the model's response may suffer a drop in accuracy. Let $\beta \in [0, 1]$ be a mixture coefficient, we will write the true joint test pdf $q_{XY}$ as a mixture of pdfs $p$ and $\upsilon$[2]:

$$q_{XY}(\boldsymbol{x}, y) = (1 - \beta) \cdot p_{XY}(\boldsymbol{x}, y) + \beta \cdot \upsilon_{XY}(\boldsymbol{x}, y). \tag{2}$$

**Remark.** *When $\beta = 0$, the test distribution matches the training distribution, i.e., there is no shift. Conversely, when $\beta = 1$, we have the largest shift between training and testing environments.*

By decomposing the joint pdfs into

$$q(X, Y) = \underbrace{Q(Y|X)}_{\text{concept}} \underbrace{q(X)}_{\text{covariate}} = q(X|Y) \underbrace{Q(Y)}_{\text{prior}}, \tag{3}$$

we can categorize three kinds of shifts that may happen. Each decomposed type of shift happens under the condition that the accompanying decomposed probability remains unchanged. Briefly, the *concept drift* is usually attributed to the presence of novel classes or concepts with covariates following the same known distribution. *Covariate shift* often happens because the input data comes from different domains, e.g., drawing of concepts while the training features are real pictures. Finally, a *prior shift* or label shift usually occurs when the test condition has a bias towards some classes. All of these shifts may have negative impacts on the model.

## 3.3 DETECTION FRAMEWORK

Predictions can be made sample by sample or window by window in a data stream.

On a **single sample** level (equivalent to OOD detection), let $s : (\boldsymbol{x}, f) \mapsto \mathbb{R}$ be a confidence-aware score function that measures how adapted the input is to the model. A low score indicates the sample is untrustworthy, and a high value indicates otherwise. This score can be simply converted to a binary detector through a threshold $\gamma \in \mathbb{R}$, i.e., $d(\cdot) = \mathbb{1}[s(\cdot, f) \leq \gamma]$. Finally, the role of the system $(d, f)$ is only to keep a prediction if the input sample $\boldsymbol{x}$ is not rejected by the detector $d$, i.e., if $d(\boldsymbol{x}) = 0$. This setup is identical to novelty, anomaly, or OOD detection. Formally, the null and alternative hypothesis writes:

$$H_0 : (X, \widehat{Y}) \sim p_{XY} \text{ and } H_A : (X, \widehat{Y}) \sim q_{XY}. \tag{4}$$

We assume that the score functions are confidence oriented, i.e., greater values indicate more confidence in prediction. So, we frame the statistical hypothesis test as a *left-tailed test* (Lehmann & Romano, 2005). Even though single-sample detection is adapted for anomaly detection, it is not well adapted for detecting distribution shifts.

In a **window based detection** scenario, we make the assumptions that 1.) there are available multiple reference samples, 2.) the instance's class label *are not* available right after prediction, and 3.) the model is not updated. So, given a *reference window* $\mathcal{W}_1^r \sim p_{XY}$ with $r$ samples and test window $\mathcal{W}_2^m = \{\boldsymbol{x}_1', \dots, \boldsymbol{x}_m'\} \sim q_X$ with sample size $m$, our task is to determine whether they are both sampled from the source distribution or, equivalently, whether $p_{XY}(\boldsymbol{x}, y)$ equals $q_{X\widehat{Y}}(\boldsymbol{x}', \hat{y}')$ where $\hat{y}' = f(\boldsymbol{x}')$. The null and alternative hypothesis of the two-sample test of homogeneity writes:

$$H_0 : p_{XY}(\boldsymbol{x}, y) = q_{X\widehat{Y}}(\boldsymbol{x}', \hat{y}') \text{ and } H_A : p_{XY}(\boldsymbol{x}, y) \neq q_{X\widehat{Y}}(\boldsymbol{x}', \hat{y}'). \tag{5}$$

In this case, the null hypothesis is that the two distributions are identical for all $(\boldsymbol{x}, y)$; the alternative is that they are not identical, which is a two-sided test. As testing this null hypothesis on a continuous and high dimensional space is unfeasible, we will compute a univariate score on each sample of the windows. With a slight abuse of notation let $s(\mathcal{W}^m, f) = \{s(\boldsymbol{x}_1, f), \dots, s(\boldsymbol{x}_m, f)\}$ be a multivariate *proxy variable* to derive a unified large-scale window-based data shift detector. To

---

[1]Also referred to in the literature as data distribution *drift*.
[2]We assume that $\upsilon$ is unknown and differs significantly from $p$, i.e., $\frac{1}{2} \int_{\mathcal{X} \times \mathcal{Y}} |p(z) - \upsilon(z)| dz \geq \delta$.

compute the final window score, we rely on the Kolmogorov-Smirnov (Massey, 1951) two-sample hypothesis test over the proxy variable. The test statistic writes:

$$\text{KS}(\mathcal{W}_1^m, \mathcal{W}_2^r) = \sup_w |F_{2,m}(w) - F_{1,r}(w)|, \tag{6}$$

where $F_{1,r}$ and $F_{2,m}$ are the empirical cumulative distribution functions (ecdf) of the scores of each sample of the first and the second widows, respectively. Finally, The KS statistic is compared to a threshold, i.e., the window-based binary detector writes $D(\cdot) = \mathbb{1}\left[\text{KS}(\cdot, \mathcal{W}_1^r) \leq \gamma\right]$.

# 4 MAIN CONTRIBUTION: ARBITRARY SCORES COMBINATION

This section explains in detail the core contribution of the paper. We present an algorithm to effectively combine arbitrary detection score functions inspired by *meta-analysis* (Glass, 1976), a statistical technique that combines the results of multiple studies to produce a single overall estimate. The first step is to transform the scores into p-values through a quantile normalization (Conover & Iman, 1981) (cf. Section 4.1). Then, with multiple detectors, the p-values can be combined using a p-value combination method (cf. Section 4.2). Finally, we introduce an additional statistical treatment, since the p-values of the multiple tests over the same sample are not independent, to obtain better-calibrated statistics through the Brown's method (Brown, 1975) (cf. Section 4.3) for the Fisher's statistic. Haroush et al. (2022) treated the first step similarly and proposed a few combination methods for the second step. However, to the best of our knowledge, we are the first to propose correcting for correlated tests in the context of OOD and data shift detection.

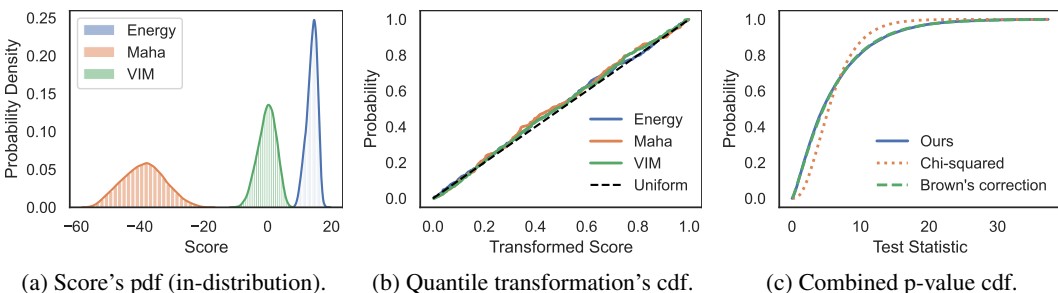

(a) Score's pdf (in-distribution).  (b) Quantile transformation's cdf.  (c) Combined p-value cdf.

Figure 1: Illustration of the three steps of the proposed algorithm on an example with three score functions on in-distribution data. Our main experiments combine 15 scores.

## 4.1 QUANTILE NORMALIZATION: MANAGING DISPARATE SCORE DISTRIBUTION

Each detector's score r.v. $S_i = s_i(X, f)$ follows very different distributions depending on the model's architecture, the dataset it was trained on, and, of course, the score function $s_i$. In order to combine them effectively, we propose to first apply a quantile transformation. Let $S_i : \Omega \mapsto \mathbb{R}$ be a continuous univariate r.v. captured by a cumulative density function (cdf) $F_i(\delta) = \Pr(S_i \leq \delta)$ for $i \in \{1, \ldots, k\}$ and $\delta \in \mathbb{R}$. Its *empirical cumulative density function* $\widehat{F}_i : \mathbb{R} \mapsto [0, 1]$ is defined by

$$\widehat{F}_i^r(\delta) = \frac{1}{r} \sum_{i=1}^r \mathbb{1}\left[S_i \leq \delta\right], \tag{7}$$

which converges almost surely to the true cdf for every $\delta$ by the Dvoretzky–Kiefer–Wolfowitz–Massart inequality (Massart, 1990). We are going to estimate this function using a subsample of size $r$ of the training or validation set if available. The resulting r.v. is uniformly distributed in the interval $[0, 1]$. As a result, for each detector $i$ and sample $\boldsymbol{x}$, we can obtain a p-value:

$$\text{p}_i(\boldsymbol{x}) = P_{H_0}\left(S_i \leq s_i(\boldsymbol{x}, f)\right) = \Pr\left(S_i \leq s_i(\boldsymbol{x}, f) \mid H_0\right) \approx \widehat{F}_i^r\left(s_i(\boldsymbol{x}, f)\right). \tag{8}$$

A decision is made by comparing the p-value to a desired significance level $\alpha$. If $\text{p} < \alpha$, then the null hypothesis $H_0$ is rejected, and the sample is believed to be OOD. Even though we derived everything for the single sample case, this formulation can be extended to the window-based scenario.

## 4.2 Combining Multiple P-Values

Our objective is to aggregate a set of $k \geq 2$ scores (or p-values) in such a way that their synthesis exhibits better properties, such as improved robustness or detection performance by consolidating each method's decision boundaries. Unfortunately, since $q$ is not known and $p$ is hard to estimate, designing an optimal test is unfeasible according to the Neyman–Pearson's Fundamental Lemma (Lehmann & Romano, 2005). However, there are several possible empirical combination methods, such as Tippett (1931) $\min_i \mathrm{p}_i$, Neyman & Pearson (1933) $2 \sum_i^k \ln(1 - \mathrm{p}_i)$, Wilkinson (1951) $\max_i \mathrm{p}_i$, Edgington (1972) $\sum_{i=1}^k \mathrm{p}_i$, and Simes (1986) $\min_i \frac{k}{i}\mathrm{p}_i$ for sorted p-values. We are going to explain in detail Fisher's method (Fisher, 1925; Mosteller & Fisher, 1948) in the main manuscript, also referred to as the chi-squared method, and Stouffer's method (Stouffer et al., 1949) in the appendix Appendix A.1, as they exhibit good properties that will be explored in the following.

If the p-values are the independent realizations of a uniform distribution, i.e., for in-distribution data, $-2 \sum_{i=1}^k \ln \mathrm{p}_i \sim \chi^2_{2k}$ follows a chi-squared distribution with $2k$ degrees of freedom. Finally, for a test input $\boldsymbol{x}$, Fisher's detector score function can be defined as

$$s_F(\boldsymbol{x}, f) = -2 \sum_{i=1}^k \ln \widehat{F}_i(s_i(\boldsymbol{x}, f)). \tag{9}$$

Fisher's test has interesting qualitative properties, such as sensitivity to the smallest p-value, and it is generally more appropriate for combining positive-valued data (Heard & Rubin-Delanchy, 2017) with matches the properties of most OOD scores.

## 4.3 Correcting for Correlated p-values

It should be noted that Fisher's method depends on the assumption of independence and uniform distribution of the p-values. However, the p-values for the same input sample are not independent. Brown (1975) proposes modeling the r.v $s_F(\cdot)$ using a scaled chi-squared distribution, i.e.,

$$s_F(\cdot) \sim c\chi^2(k'), \quad \text{with} \quad c = \mathrm{Var}(S_F)/(2\mathbb{E}[S_F]) \quad \text{and} \quad k' = 2(\mathbb{E}[S_F])^2/\mathrm{Var}(S_F). \tag{10}$$

With this simple trick, we approach more interpretable results, as we know in advance the distribution followed by the in-distribution data under our combined score. As so, we can leverage calibrated confidence values given by the true cdf and leverage more powerful single-sample statistical tests for window-based data shift detection.

**Remark.** *Commonly, the binary detection threshold $\gamma$ for a score is set based on a certain quantile of the score's value on an in-distribution validation set. Usually, this value is set to have 95% of entities correctly classified. By combining p-values with Fisher's method and correcting for correlation with Brown's method, we relax the need of a validation set to find $\gamma$, i.e., $\gamma = F_{c\chi^2(k')}^{-1}(\alpha)$.*

## 5 Experimental Setup

In this section, we present and detail the experimental setup from a conceptual point of view. For all our main experiments, we set as *in-distribution* dataset *ImageNet-1K* (=ILSVRC2012; Deng et al., 2009) on ResNet (He et al., 2016) and Vision Transformers (Dosovitskiy et al., 2021) models. Our experiments encompass a full-spectrum setting on i.) classic OOD detection (Section 5.1), ii.) concept shift via independent window-based detection (Section 5.2; Par. 1), iii.) covariate shift via independent window-based detection (Section 5.2; Par. 2), and iv.) sequential shift detection via sequential window-based detection (Section 5.3).

## 5.1 Classic Out-of-Distribution Detection

We evaluate OOD detection performance on the curated **datasets** from Bitterwolf et al. (2023) that contain a clean subset of the far-OOD datasets: SSB-Easy (Vaze et al., 2022), OpenImage-O (Wang et al., 2022), Places (Zhou et al., 2017), iNaturalist (Horn et al., 2017), and Textures (Cimpoi et al., 2014); and the near-OOD datasets: SSB-Hard (Vaze et al., 2022), Species (Hendrycks et al., 2022), and NINCO (Bitterwolf et al., 2023). For the **evaluation metrics**, we consider the Area Under

the Receiver Operating Characteristic curve (AUROC), which measures how well the OOD score distinguishes between out- and in-distribution data in a threshold-independent manner (higher is better). For the **baselines**, we consider the following post-hoc detection methods: MSP (Hendrycks & Gimpel, 2017), Energy (Liu et al., 2020), Maha (Lee et al., 2018), Igeood (Gomes et al., 2022), MaxCos (Techapanurak et al., 2020), ReAct (Sun et al., 2021), ODIN (Liang et al., 2018), DICE (Sun & Li, 2022), VIM (Wang et al., 2022), KL-M (Hendrycks et al., 2022), Doctor (Granese et al., 2021), RMD (Ren et al., 2021), KNN (Sun et al., 2022b), GradN (Huang et al., 2021). When needed, we followed the hyperparameter selection procedure suggested in the original papers. New methods can be easily integrated into our universal framework and should improve the robustness and, potentially, the performance of the group detector.

## 5.2 INDEPENDENT WINDOW-BASED DETECTION

**Concept shift.** We suppose that full ID and corrupted windows formed by ID and OOD data from the OpenImage-O (OI-O) (Wang et al., 2022) dataset with mixing parameter $\beta$ (Equation (2)) are available. The objective of the detectors is to classify each test window as being corrupted or not by comparing it to a fixed reference window of size $r = 1000$ extracted from a validation set. We ran experiments with $\beta \in [0, 1]$ and with window sizes $|\mathcal{W}| \in \{1, \ldots, 1000\}$. We use the KS two sample test described in Section 3.3 as window-based test statistics. Evaluation metrics and baselines are the same as described in Section 5.1. Figure 2 shows Fisher's ensembled test statistic in different scenarios of mixture amount and window sizes. Figure 2a shows the distribution of the test statistics for different mixture values from $\beta = 0$ (fully ID window) to $\beta = 1$ (fully OOD window). Figure 2b displays how the distribution on the test statistic changes from flatter to peaky as we increase the window size (better seen in color). Finally, Figure 2c demonstrates how the detection performance is affected by window sizes increase mixture coefficient. Note an AUROC of 0.5 for the case with $\beta = 0$, as expected. With a window size as low as 8, we can already perfectly distinguish fully corrupted from normal ones. Similar qualitative behavior is observed on all detectors.

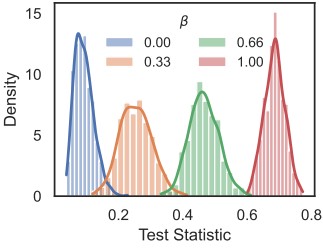 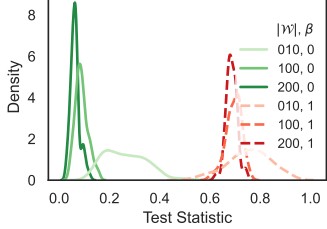 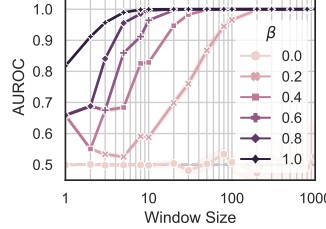

(a) From ID to OOD window.  (b) Flat to peaky windows.  (c) Detection performance.

Figure 2: Test statistic distributional behavior and detection performance as a function of the concept shift intensity and window size. Experiments ran for Fisher's method on a ResNet-50.

**Covariate shift.** We ran experiments with the ImageNet-R (IN-R) (Hendrycks et al., 2021) dataset providing domain shift to 200 ID classes. Similarly to the novelty setup described in the previous paragraph, we suppose that the windows arrive independently from one another. We use the same reference window to compute metrics and we vary the mix parameter and the window size in the same way. Figure 8 is similar to Figure 2 and shows the behavior of the combined p-values for detecting covariate shift in windows of a data stream. Similar qualitative observations are drawn. Table 1 display the accuracy of each model studied

| Model | Train | Val. | IN-R | IN-R (m) |
|-------|-------|------|------|----------|
| RN-50 | 87.5 | 76.1 | 1.33 | 36.2 |
| RN-101 | 90.0 | 77.4 | 1.67 | 39.3 |
| RN-152 | 90.2 | 78.3 | 0.67 | 41.4 |
| ViT-S-16 | 88.0 | 81.4 | 1.33 | 46.0 |
| ViT-B-16 | 90.5 | 84.5 | 3.33 | 56.8 |
| ViT-L-16 | 92.3 | 85.8 | 1.67 | 64.3 |

Table 1: Top-1 accuracies in percentage on the training and validation sets and on the domain drift on all and (m)asked classes outputs.

on the new domain. We can see that without masking only the classes present on IN-R, the drift is severe, with a top-1 accuracy of around 1% only. However, as we compute the top accuracy only on the 200 classes by masking the other 800, we can observe an amelioration in performance. In our experiments, we simulate the more realistic scenario by supposing that this mask is not available.

## 5.3 SEQUENTIAL DRIFT DETECTION

In this setup, differently from the independent window-based detection setting, we implement a sliding window of size 64 with a stride of one. We assume that the samples arrive sequentially and labels are unavailable to compute the true accuracy of the model on the current or past test windows. The objective is to see how well the moving average of the detection score will correlate with the moving accuracy of the model. By having a high correlation with accuracy, a machine learning practitioner can use the values of the score as an indicator if the system is suffering from any degrading data distribution shift. We ran experiments with the corrupted ImageNet (IN-C) (Hendrycks & Dietterich, 2019) dataset. The intensity of the drift increases over time from in-

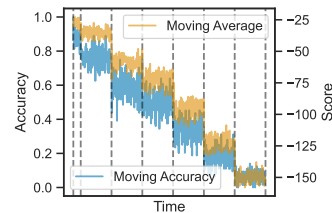

Figure 3: Data stream monitoring with correlation $\rho = 0.98$.

tensity 0 (training warmup set and part of the validation set without corruptions) to 5. Figure 3 illustrates the monitoring pipeline with the moving accuracy on the left y-axis and the score's moving average on the right y-axis. The score's moving average can effectively follow the accuracy (hidden variable).

## 6 RESULTS AND DISCUSSION

**Out-of-distribution Detection.** Table 2 displays the experimental result on classic OOD detection for a ResNet-50 model on the setup described in Section 5.1. Fisher's method achieves state-of-the-art results on average AUROC, surpassing the previous SOTA by 1.4% (MaxCos). Also, the other six standard p-value combination strategies also achieve great results, validating our proposed meta-framework of Section 4. Similar tables for FPR and other architectures are available in the Appendix A. Apart from achieving overall great performance capabilities, the most compelling observed property is the robustness compared to individual detection metrics. Figure 4 shows the ranking per dataset and on average for selected methods. We can observe that, even though several detectors achieve top-1 performance in a few cases, there are several datasets in which they underperform, sometimes catastrophically. This is not true for the group methods, which can effectively combine the existing detectors to obtain a final score that successfully combines the multiple decision regions, keeping top-4 performance in all cases (Fisher).

Table 2: Numerical results in terms of AUROC (values in percentage) comparing p-value combination methods against literature for a ResNet-50 model trained on ImageNet. The left-hand side shows results on out-of-distribution detection and the right-hand side shows results on concept (OI-O) and covariate (IN-R) shift detection with $|\mathcal{W}| = 3$ and $\beta = 1$.

| Method | Avg. | SSB-H | NINCO | Spec. | SSB-E | OI-O | Places | iNat. | Text. | IN-R | OI-O |
|---|---|---|---|---|---|---|---|---|---|---|---|
| | | | | Out-of-Distribution Detection | | | | | | Data Shift Detection | |
| Fisher | **89.8** | 75.8 | 84.3 | 88.7 | 91.0 | **93.0** | 93.1 | 95.9 | 96.4 | **94.3** (0.2) | **95.7** (0.4) |
| Stouffer | 89.6 | 75.5 | 84.6 | 89.0 | 90.9 | 92.8 | 92.7 | 95.8 | 95.5 | 92.8 (0.2) | 95.5 (0.4) |
| Edgington | 89.3 | 75.2 | 84.6 | 89.0 | **91.0** | 92.5 | 92.1 | 95.5 | 94.4 | 92.5 (0.2) | 95.3 (0.3) |
| Pearson | 89.2 | 74.6 | **84.9** | **89.4** | 90.9 | 92.4 | 91.8 | 95.5 | 94.1 | 92.2 (0.3) | 93.9 (0.4) |
| Simes | 89.2 | 75.0 | 83.0 | 87.6 | 89.5 | 92.3 | 93.1 | 95.7 | 97.0 | 83.6 (0.5) | 86.6 (0.7) |
| Tippet | 88.5 | 74.8 | 80.9 | 86.7 | 87.3 | 91.7 | 93.5 | 95.9 | 97.2 | 82.0 (1.0) | 81.5 (0.7) |
| Wilkinson | 86.5 | 68.7 | 83.3 | 89.0 | 88.1 | 89.5 | 86.3 | 93.6 | 93.1 | 71.2 (1.8) | 77.4 (0.9) |
| MaxCos | 88.4 | 69.6 | 82.7 | 88.2 | 89.9 | 92.2 | 89.7 | 96.1 | **98.4** | 92.2 (0.3) | 95.5 (0.4) |
| ReAct | 87.4 | 75.0 | 80.1 | 87.2 | 82.3 | 90.4 | **95.8** | **96.6** | 91.6 | 92.2 (0.3) | 94.5 (0.4) |
| ODIN | 85.4 | 72.9 | 80.3 | 83.9 | 87.7 | 88.8 | 90.0 | 91.4 | 88.3 | 92.2 (0.5) | 93.6 (0.4) |
| DICE | 85.1 | 70.2 | 77.4 | 84.1 | 82.5 | 88.6 | 91.6 | 94.4 | 91.9 | 85.5 (0.3) | 90.1 (0.4) |
| Energy | 85.0 | 72.1 | 79.6 | 83.1 | 87.2 | 88.7 | 90.0 | 90.7 | 88.4 | 91.9 (0.3) | 93.4 (0.4) |
| Igeood | 84.7 | 71.4 | 80.1 | 83.0 | 88.8 | 88.0 | 88.8 | 90.2 | 87.6 | 91.0 (0.3) | 93.3 (0.3) |
| VIM | 84.3 | 66.4 | 78.9 | 80.7 | 89.3 | 90.3 | 83.7 | 87.9 | 97.5 | 92.2 (0.5) | 95.4 (0.4) |
| KL-M | 84.3 | 73.9 | 80.7 | 86.1 | 87.3 | 85.7 | 85.2 | 90.0 | 85.3 | 86.9 (0.6) | 91.4 (0.9) |
| Doctor | 84.2 | 75.9 | 80.6 | 85.1 | 87.0 | 85.1 | 86.7 | 89.7 | 83.8 | 85.2 (0.6) | 89.9 (0.4) |
| RMD | 83.5 | **78.2** | 82.7 | 87.7 | 82.9 | 84.9 | 81.3 | 87.6 | 82.7 | 89.9 (0.3) | 93.1 (0.6) |
| MSP | 83.5 | 75.5 | 79.9 | 84.5 | 86.1 | 84.1 | 85.9 | 88.7 | 83.0 | 83.6 (0.5) | 89.0 (0.4) |
| KNN | 83.4 | 64.3 | 79.6 | 83.3 | 88.0 | 87.2 | 83.0 | 84.1 | 97.6 | 84.6 (0.5) | 89.2 (0.8) |
| GradN | 82.6 | 63.3 | 74.4 | 83.1 | 76.2 | 84.4 | 91.1 | 96.0 | 92.5 | 49.7 (1.0) | 67.4 (1.2) |
| Maha | 69.6 | 55.3 | 65.7 | 70.3 | 70.6 | 73.9 | 60.0 | 72.7 | 88.4 | 71.2 (1.8) | 77.6 (1.8) |

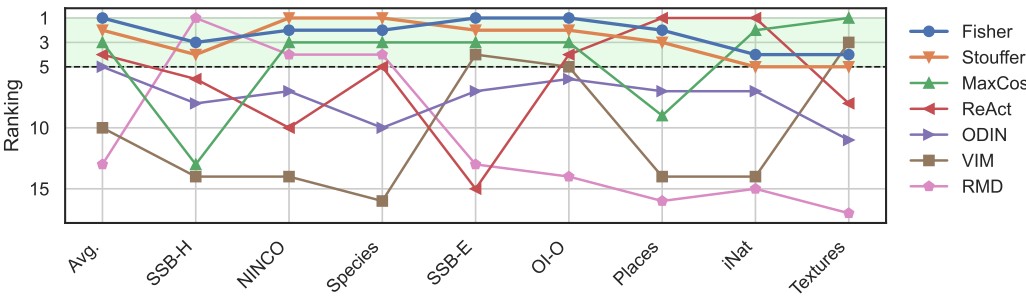

Figure 4: Ranking in terms of AUROC for a few selected methods for the ResNet-50 model. Note that the two displayed methods to combining tests obtain a top-5 ranking in every dataset, while state-of-the-art individual detectors vary significantly in performance.

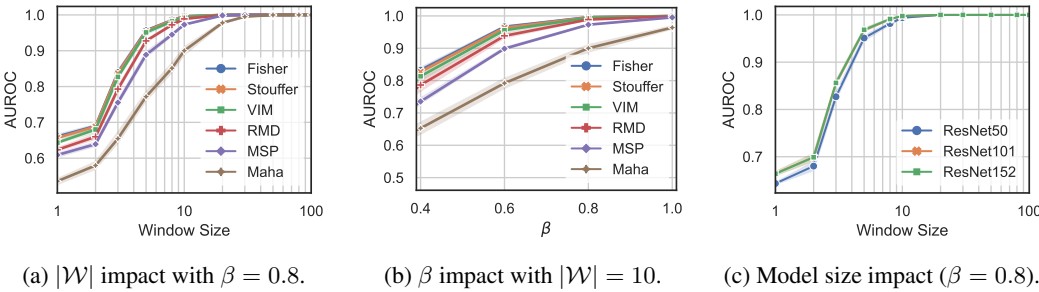

(a) $|\mathcal{W}|$ impact with $\beta = 0.8$.     (b) $\beta$ impact with $|\mathcal{W}| = 10$.     (c) Model size impact ($\beta = 0.8$).

Figure 5: Concept shift (OpenImage-O) detection performance on a ResNet-50 model (ImageNet).

**Independent Window-Based Detection.** Figure 5 displays results on concept shift detection. Figure 5a) shows the detectors' performance with the window size, showcasing a small edge in performance for Vim, Fisher's, and Stouffer's methods. Figure 5b displays the impact of the mixture parameter. Figure 5c shows that model size does mildly impact detection performance, with registered improvements for ResNet-152 over ResNet-50 on Fisher's method. The confidence interval bounds are computed over 10 different seeds and are quite narrow for all methods. Similar observations are drawn in the covariate shift results displayed in Figure 10, except for the network scale impact, where we obtained more or less the same results for all sizes. On the right-hand side of Table 2, we showed that for both shifts, we demonstrated improved performance by combining p-values, especially with Fisher's method. We also observe from the table that the concept shift benchmark is slightly easier than the covariate shift benchmark, probably biased because most OOD detectors were developed for the novel class scenario. Additional results are available in the Appendix A.

**Results in a sequential stream.** Table 3 displays the average results for the ImageNet-C dataset, including 19 kinds of covariate drifts. We can observe that the most performing methods are the scores function based on the softmax and logit outputs and that Fisher's method is on par with top-performing methods. We emphasize that, even though MSP and Doctor works well in this benchmark, they demonstrated poor performance on other benchmarks, notably on Table 2. This supports our claim that combining scores is the most effective approach for improving robustness and performance in general data shift detection.

Table 3: Average Pearson's correlation coefficient with the hidden accuracy with one standard deviation in parenthesis for top and bottom performing detection methods across 19 different corruptions on the sequential data shift detection scenario on a ResNet-50 model.

| | Fisher | Doctor | MSP | Igeood | ... | KNN | RMD | GradN | Maha |
|---|---|---|---|---|---|---|---|---|---|
| Avg. | 0.96 (0.03) | 0.96 (0.03) | 0.96 (0.03) | 0.95 (0.03) | ... | 0.92 (0.07) | 0.92 (0.03) | 0.91 (0.07) | 0.81 (0.21) |

**On the distillation of the best subset of detectors.** We provide a supervised study to showcase the potential impact of finding an optimal subset of detectors. We computed the performance of all possible subsets of $j < k$ methods, and we report our results in Figure 6. We found out that 1.) surprisingly, removing the least performant detector from the pool does not necessarily increase performance; 2.) increasing the size of the subset improves probable detection on average and on worst performance; 3.) best subset selection benefits harder to find OOD samples; and 4.) not surprisingly, the best combination for the easy benchmark may be very different from the best subset on the harder one. We also list the best subset of four methods on average performance: {GradN, ReAct, MaxCos, RMD}, on an easy dataset (SSB-Easy): {DICE, MaxCos, KL-M, VIM}, and on a hard dataset (SSB-Hard): {MSP, GradN, ReAct, RMD}. Their AUROC and relative gain w.r.t all methods combined together are equal to 91.4 (+1.8%), 92.0 (+1.1)%, and 79.7 (+4.9%), respectively. *These observations support the main claim of the paper that in a data-free scenario with specialized methods, combining all of them should greatly improve the safety of the underlying system.*

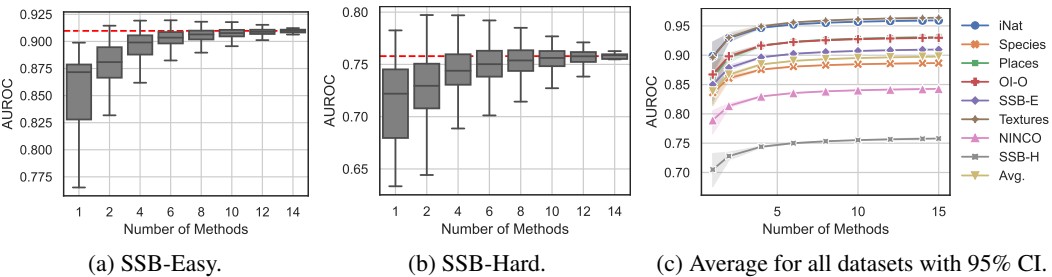

|  (a) SSB-Easy. | (b) SSB-Hard. | (c) Average for all datasets with 95% CI. |

Figure 6: Evaluation of all possible subsets of detectors on the OOD detection benchmark. The dashed red line indicates the performance combining all detectors.

**Limitations.** Our study acknowledges that there not a one-size-fits-all detector or a universally superior combination method, a finding supported by previous research (Heard & Rubin-Delanchy, 2017; Fang et al., 2022). This recognition underlines the inherent complexity of real-world ML applications. Additionally, we recognize that the empirical cumulative distribution function may be susceptible to estimation errors, and the effectiveness of individual detector score functions can influence the performance of the aggregated score. It is also important to note that, although our investigation primarily focused on computer vision applications, similar techniques can be applied to diverse scenarios and application domains.

**Future Directions.** We believe several directions for future research are left open. A promising path involves exploring the pattern in the performance of detectors across different kinds of drifts to enable subset selection, leading to enhanced detection accuracy. However, it might need validation on held-out labeled data or domain expertise to reflect the prior importance of the p-values. Furthermore, our proposed algorithm could be integrated into incremental and online learning algorithms, thereby enhancing their adaptability to evolving data streams, representing an exciting avenue for advancing machine learning applications.

## 7    CONCLUSION

This paper introduces a highly adaptable and efficient approach to combining detectors while effectively addressing data distribution shifts. By converting arbitrary scores into p-values and incorporating meta-analysis tools, we have demonstrated consolidated decision boundaries that prevent catastrophic collapses observed on individual detectors. We also showed that Fisher's method corrected for correlated p-values demonstrates great properties, being a fully interpretable detection criterion. Through a meticulous empirical investigation, we have thoroughly validated our approach, assessing both single-example out-of-distribution detection and window-based data distribution shift detection, gaining significant robustness and detection performance across various domains. Looking ahead, our framework offers a robust foundation for enhancing the safety of AI systems.

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

# A APPENDIX

## A.1 COMBINING MULTIPLE P-VALUES WITH STOUFFER'S METHOD

The Stouffer et al. (1949) test statistics for combining p-values is given by:

$$s_S(\cdot) = \sum_{i=1}^{k} \Phi^{-1}(\mathrm{p}_i(\cdot)) \tag{11}$$

where $\Phi^{-1}$ is the *probit*, i.e., $\Phi^{-1}(\alpha) = \sqrt{2}\,\mathrm{erf}^{-1}(2\alpha - 1)$, where $\mathrm{erf}$ is the Gauss error function. If the p-values are independent, $s_S(\cdot) \sim \mathcal{N}(0, 1)$, where $\mathcal{N}(\mu, \sigma^2)$ is the normal distribution with mean $\mu$ and standard deviation $\sigma$.

## A.2 CORRECTING FOR CORRELATED P-VALUES WITH HARTUNG'S METHOD

Hartung (1999) method aims to correct Stouffer's test for correlated p-values. The group statistics writes:

$$s_H(\cdot; \boldsymbol{w}, \rho) = \frac{\sum_{i=1}^{k} w_i \Phi^{-1}(\mathrm{p}_i(\cdot))}{\sqrt{(1 - \rho)\sum_{i=1}^{k} w_i^2 + \rho\left(\sum_{i=1}^{k} w_i\right)^2}} \underset{H_0}{\sim} \mathcal{N}(0, 1) \tag{12}$$

with $\rho$ a real-valued parameter and $\sum_{i=1}^{k} w_i \neq 0$. Hartung showed that an unbiased estimator of $\rho$ based on $\mathrm{p}_i$ under $H_0$ is given by:

$$\hat{\rho} = 1 - \mathbb{E}\left[\frac{1}{k-1}\sum_{i=1}^{k}\left(\Phi^{-1}(\mathrm{p}_i) - \frac{1}{k}\sum_{i=1}^{k}\Phi^{-1}(\mathrm{p}_i)\right)^2\right]. \tag{13}$$

Assuming equal weights, we repeated a similar experiment as the one of Figure 1, replacing the chi-squared with a standard normal to see how well the correction works. We can observe in Figure 7 that the corrected statistic indeed approximates a standard normal distribution. Unlike Brown's method, Hartung's method corrects the statistics directly instead of correcting the parameters of the underlying distribution.

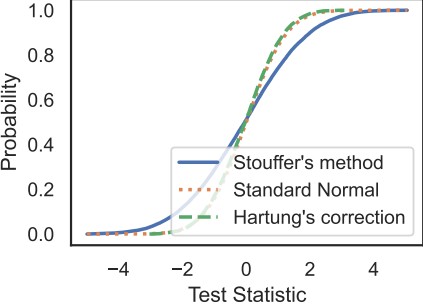

Figure 7: Stouffer's method corrected for correlated p-values with Hartung's method to obtain a standard normal distribution when evaluated on in-distribution data (null hypothesis), also obtaining interpretable results.

## A.3 ADDITIONAL PLOTS

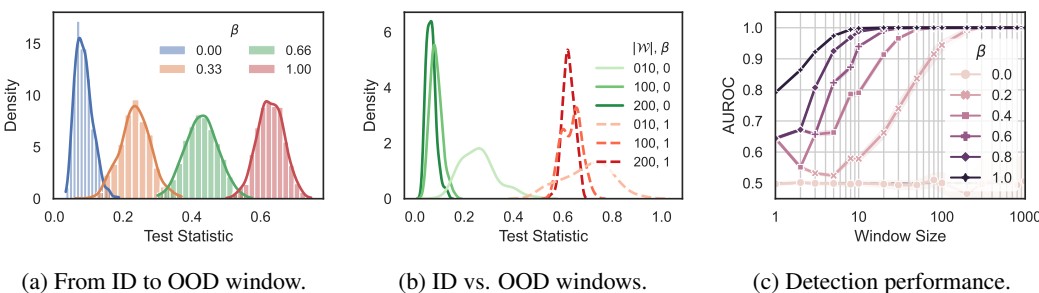

(a) From ID to OOD window.  (b) ID vs. OOD windows.  (c) Detection performance.

Figure 8: Test statistic behavior and detection performance in function of the covariate shift intensity and window size. Experiments ran on a ResNet-50.

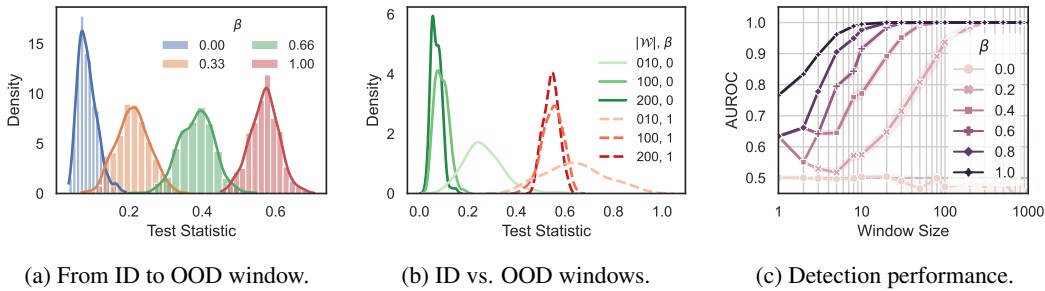

(a) From ID to OOD window.  (b) ID vs. OOD windows.  (c) Detection performance.

Figure 9: Test statistic behavior and detection performance in function of the covariate shift intensity and window size. Experiments ran on a ViT-L-16.

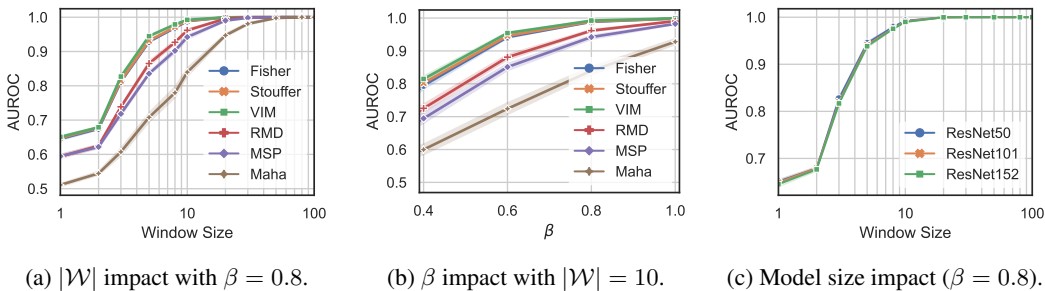

(a) $|\mathcal{W}|$ impact with $\beta = 0.8$.  (b) $\beta$ impact with $|\mathcal{W}| = 10$.  (c) Model size impact ($\beta = 0.8$).

Figure 10: Covariate shift (ImageNet-R) detection performance on a ResNet-50 model (ImageNet).

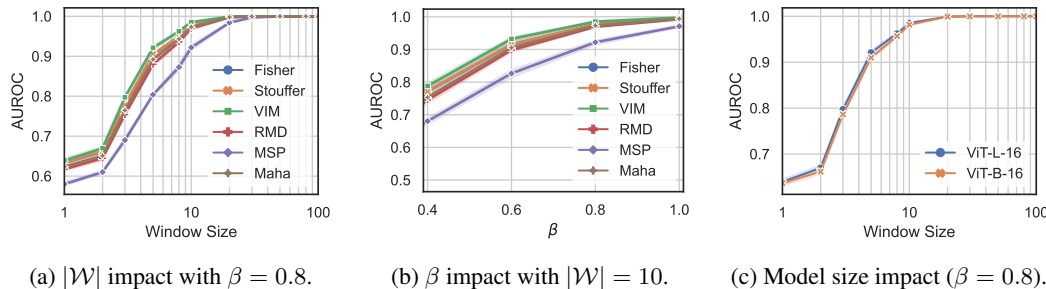

(a) $|\mathcal{W}|$ impact with $\beta = 0.8$.    (b) $\beta$ impact with $|\mathcal{W}| = 10$.    (c) Model size impact ($\beta = 0.8$).

Figure 11: Covariate shift (ImageNet-R) detection performance on a ViT-L-16 model (ImageNet).

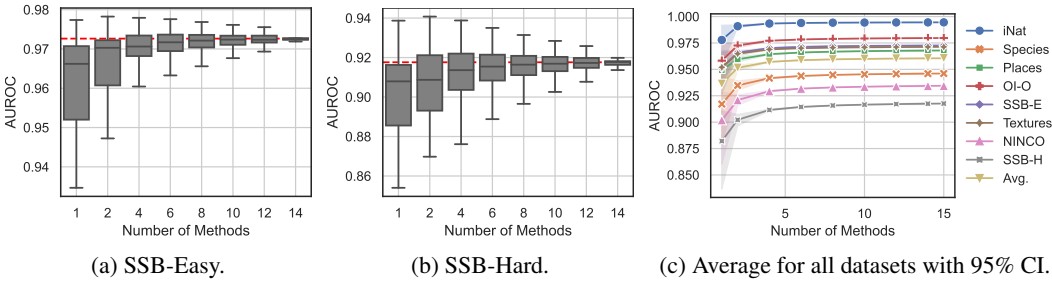

(a) SSB-Easy.    (b) SSB-Hard.    (c) Average for all datasets with 95% CI.

Figure 12: Evaluation of all possible subsets of detectors on the OOD detection benchmark for a ViT-L-16 model. The dashed red line indicates the performance combining all detectors.

## A.4 ADDITIONAL TABLES

Table 4: Numerical results in terms of AUROC (values in percentage) comparing p-value combination methods against literature for a ViT-L-16 model trained on ImageNet.

| Method | Avg. | SSB-H | NINCO | Spec. | SSB-E | OI-O | Places | iNat. | Text. |
|---|---|---|---|---|---|---|---|---|---|
| Maha | **96.8** | 92.7 | **94.8** | 96.6 | 97.4 | **98.6** | 96.9 | **99.8** | 97.6 |
| VIM | 96.6 | 92.1 | 93.9 | 95.6 | **97.7** | 98.5 | 96.7 | 99.7 | **98.2** |
| RMD | 96.1 | 92.4 | **94.8** | 96.2 | 96.3 | 97.9 | 95.7 | 99.5 | 95.6 |
| Fisher | 96.1 | 91.8 | 93.4 | 94.6 | 97.3 | 98.0 | 96.8 | 99.5 | 97.1 |
| Vovk | 96.1 | 91.8 | 93.4 | 94.6 | 97.3 | 98.0 | 96.8 | 99.5 | 97.1 |
| Simes | 96.0 | 91.7 | 93.4 | 94.6 | 97.1 | 98.0 | **97.0** | 99.5 | 97.0 |
| Stouffer | 96.0 | 91.5 | 93.3 | 94.4 | 97.3 | 97.9 | 96.7 | 99.4 | 97.1 |
| ReAct | 95.9 | **93.9** | 94.7 | **96.9** | 96.6 | 97.8 | 91.1 | 99.5 | 96.3 |
| Edgington | 95.7 | 90.9 | 92.8 | 93.9 | 97.1 | 97.7 | 96.8 | 99.2 | 97.1 |
| Energy | 95.6 | 91.0 | 92.5 | 93.2 | 97.3 | 97.8 | 96.4 | 99.3 | 97.1 |
| Tippet | 95.5 | 90.9 | 92.3 | 94.6 | 96.4 | 97.6 | 96.9 | 99.3 | 96.2 |
| Pearson | 95.5 | 90.4 | 92.4 | 93.6 | 97.1 | 97.6 | 96.8 | 99.0 | 97.0 |
| MaxL | 95.5 | 91.2 | 92.6 | 93.2 | 97.0 | 97.6 | 96.1 | 99.3 | 96.8 |
| ODIN | 95.5 | 91.2 | 92.6 | 93.2 | 97.0 | 97.6 | 96.1 | 99.3 | 96.8 |
| Igeood | 95.4 | 90.8 | 92.6 | 93.2 | 97.1 | 97.6 | 96.0 | 99.2 | 96.7 |
| MaxCos | 94.9 | 89.7 | 91.2 | 92.9 | 97.0 | 96.9 | 96.2 | 98.2 | 97.1 |
| GradN | 94.9 | 90.1 | 91.4 | 91.8 | 96.6 | 97.3 | 96.1 | 99.2 | 96.3 |
| KNN | 93.4 | 85.4 | 89.2 | 91.9 | 96.3 | 96.1 | 94.3 | 97.6 | 96.4 |
| Doctor | 93.1 | 88.9 | 90.3 | 91.8 | 94.1 | 94.8 | 93.2 | 98.4 | 93.7 |
| MSP | 92.5 | 88.2 | 89.5 | 91.3 | 93.5 | 94.0 | 92.4 | 98.0 | 93.0 |
| KL-M | 92.1 | 85.4 | 89.0 | 90.6 | 93.5 | 94.2 | 92.5 | 98.0 | 93.7 |
| Wilkinson | 91.2 | 81.6 | 85.0 | 87.1 | 94.2 | 94.7 | 96.3 | 95.7 | 95.2 |
| DICE | 76.3 | 60.2 | 63.6 | 67.0 | 79.8 | 80.8 | 94.3 | 81.9 | 82.5 |

