# OpenReview forum: "Combine and Conquer: A Meta-Analysis on Data Shift and Out-of-Distribution Detection"
_ICLR.cc/2024/Conference — ICLR 2024 Conference Withdrawn Submission_

### Official Review · Reviewer_yLhF · 2023-11-01

**Soundness:** 3 good
**Presentation:** 1 poor
**Contribution:** 2 fair
**Rating:** 5
**Confidence:** 3

**Summary:**

The paper presents a universal method for integrating detectors and evaluating approaches for detecting out-of-distribution (OOD) data and addressing data distribution shifts. The authors propose a technique that normalizes detector scores into p-values using quantile normalization, transforming the problem into a multivariate hypothesis test. They combine these tests using meta-analysis tools, improving the effectiveness of the detector and consolidating decision boundaries. The authors also create an interpretable criterion by adjusting the final statistics of in-distribution scores. Through empirical investigation, they demonstrate that their approach enhances robustness and performance across various domains, shift types, and OOD detection scenarios. The paper contributes to the field of machine learning by providing a flexible framework for integrating detectors and addressing OOD detection challenges.

**Strengths:**

The paper presents a universal method for integrating detectors, which can be applicable across different domains, including OOD detection and two sample test.

The use of quantile normalization to transform detector scores into p-values seems interesting. This transformation allows for treating the problem as a multivariate hypothesis test and enables effective combination for a set of predefined scoring strategies.

The paper proposes adjusting the final statistics of in-distribution scores to create a fully interpretable criterion. This feature is valuable as it provides insights and explanations for the detection decisions, enhancing the transparency and interpretability of the method.

Through empirical investigation, the paper demonstrates that the proposed method significantly enhances overall robustness and performance across various domains, shift types, and out-of-distribution detection scenarios. This finding highlights the practical effectiveness of the approach.

**Weaknesses:**

This paper mainly uses the scoring strategies from OOD detection to build their method, which mainly considers the concept shift. However, in the discussion, they actually present another two different distribution shift, i.e., covariate shift and prior shift. Then, a natural question is why the basic method that handle concept shift can also be useful to tackling covariate shift and prior shift? Is it the attributed to the combination of different p values? More detailed discussion and empirical evaluation seem to be important.

Quantile normalization seems interesting, while there exist many other ways, typically more simple ways, in doing normalisation. For example, we can simply normalise the score to follow the N(0,1) Gaussian distribution, which can also normalise the data. Therefore, more discussion about the theoretical superiority in using quantile normalisation is interesting.

It seems that extra parameters are introduced when combine different scoring strategies in Section 4.2 and 4.3, while I cannot find how to tune the related parameters. Moreover, how to conduct hyper parameter tuning, epically for the choice of evaluation dataset, should be discussed. If the proposed method does not introduce additional parameters, seemingly different scoring strategies are treated equally in combination. I am not sure if it is a proper setup.

Different scoring strategies are effective for varying data, so another interesting question is that if the proposed method can be instance dependent in combination when facing different data points.

**Questions:**

Please see the weaknesses above.

---

### Official Review · Reviewer_UAHM · 2023-11-01

**Soundness:** 2 fair
**Presentation:** 3 good
**Contribution:** 2 fair
**Rating:** 3
**Confidence:** 4

**Summary:**

The paper presents an approach to combine multiple arbitrary out-of-distribution (OOD) detection scores. In particular, quantile normalisation is applied to obtain p-values from the individual detectors. Then, scores from multiple OOD detectors are combined using Fisher or Stouffer meta-analysis to obtain a single OOD estimate. As Fisher's method assumes independence of the p-values, a Brown correction is proposed, using the scaled chi-squared distribution. The approach is evaluated on standard OOD detection benchmarks where it shows good performance compared to the prior work.

**Strengths:**

+ The idea of the paper is easy to follow. In addition, the problem is well presented, including a detailed explanation of the different types of data shifts.
 + The different single-score methods and methods combining different detection scores are evaluated in detail using different data shift scenarios.

**Weaknesses:**

- (Major) Although the results of combining existing methods are interesting, the paper does not show any new idea. It therefore has limited novelty.

- (Major) The paper claims that a major contribution is the correction for the assumed independence of Fisher's method. Therefore, Fisher's method should be compared with and without the correction in the experiments.

- (Major) It is claimed that the method is interpretable as the distribution of the combined scores is known in advance. However, this is not demonstrated in the paper.

- The clarity and notation of the method has room for improvement, e.g. in Section 4.1 the index i is used to iterate over the window examples and at the same time the doctor.

- The experimental protocol is not very clear. For example, it should be clearly stated which combination of detectors is used in the evaluation. Is this adjusted according to the shift or is it sample or window based?

**Questions:**

- The paper states that different detectors are ensembled. They have a study on the distillation of the best subset of detectors where different OOD detection scores are combined. Are different classifiers combined or are different OOD detection scores combined? So does the word "detector" refer to the different OOD detection scoring methods or does it refer to a classification model? This is not immediately clear.

- It should be clarified how the indices in equation 6 are mixed up? Should they be W^r_1 and W^m_2?

---

### Official Review · Reviewer_2NKy · 2023-11-03

**Soundness:** 3 good
**Presentation:** 3 good
**Contribution:** 1 poor
**Rating:** 6
**Confidence:** 2

**Summary:**

The paper addresses the challenge of detecting distribution shifts in data streams that are inputted to deep neural networks. It emphasizes the importance of recognizing when the distribution of incoming data deviates from the distribution of the training data, which can impact the performance and reliability of the model.

**Strengths:**

1, Instead of instance-level discrimination on OOD samples, this paper considers the OOD detection from an interesting perspective: the windows from the streamed data.
2, In the poposed detection framework, the author leverages empirical cumulative distribution functions yo effectively compare the distribtuion from two windows, reference window and the test one.
3.  The transformation into p-values is reasonable, and the calibration across detectors is well motivated.
4. Extensive experiments are conducted to verify the effectiveness of the proposed method.

**Weaknesses:**

This paper falls outside of my expertise sligtly, thus for now, I cannot find a clear weaknesses.

**Questions:**

1, In Fig 5 (c), I cannot find the curve correpsonding to Resnet-101.
2, For the ablation study on window sizes, will listing the proportion of window size towards the whole dataset helpful for understanding the impact of window sizes?

**Details Of Ethics Concerns:**

None.